# Influence of Sex and Age on Site of Onset, Morphology, and Site of Metastasis in Colorectal Cancer: A Population-Based Study on Data from Four Italian Cancer Registries

**DOI:** 10.3390/cancers15030803

**Published:** 2023-01-28

**Authors:** Viviana Perotti, Sabrina Fabiano, Paolo Contiero, Maria Michiara, Antonio Musolino, Lorenza Boschetti, Giuseppe Cascone, Maurizio Castelli, Giovanna Tagliabue

**Affiliations:** 1Cancer Registry Unit, Fondazione IRCCS, Istituto Nazionale dei Tumori, 20133 Milan, Italy; 2Environmental Epidemiology Unit, Fondazione IRCCS, Istituto Nazionale dei Tumori, 20133 Milan, Italy; 3Department of Medicine and Surgery, University of Parma, Medical Oncology, Cancer Registry, University Hospital of Parma, 43100 Parma, Italy; 4Epidemiology Unit, Health Protection Agency of Pavia (ATS Pavia), 27100 Pavia, Italy; 5Ragusa Cancer Registry, Department of Prevention, Ragusa Health Authority, 97100 Ragusa, Italy; 6Cancer Registry, Aosta Valley Health Authorities Department of Public Health, 11100 Aosta, Italy

**Keywords:** colorectal cancer, population-based, cancer registry, metastasis, age, sex, relative survival

## Abstract

**Simple Summary:**

Colorectal cancer has the third highest cancer incidence in the world. The purpose of our population-based study was to evaluate how non-modifiable factors (sex and age) influence the biological and clinical characteristics of the disease, especially in patients with metastases at diagnosis. Our results indicate that sex and age influence the site of onset, morphology, and metastatic pattern of colorectal cancer, thereby affecting patient survival. Understanding the biological mechanisms that are regulated by sex and age, such as the hormonal profile, may contribute to the earlier diagnosis and better prognosis of this disease. As colorectal cancer is subject to screening, the influence of sex and age should be taken into account to implement a targeted strategy that considers these two factors in the general population.

**Abstract:**

The prognosis of colorectal cancer is affected by factors such as site of origin, tumor morphology, and metastasis at diagnosis, but also age and sex seem to play a role. This study aimed to investigate within the Italian population how sex and age interact in influencing certain aspects of the disease and how they affect patient survival, particularly in the metastatic cohort. Data from four cancer registries were collected, and patients were classified by sex and age (<50, 50–69, and >69 years). Two separate analyses were conducted: one for patients having right or left colon cancer with adenocarcinoma or mucinous morphology, and one for patients having metastases at diagnosis. Women showed significant differences in right colon cases from the youngest to oldest age group (36% vs. 45% vs. 60%). Men <50 years had a significantly higher mucinous carcinoma percentage than their female counterparts (22% vs. 11%), while in the oldest age group women had the highest percentage (15% vs. 11%). The metastatic pattern differed between men and women and by age. The three-year relative survival in the <50 age group was better for women than men, but this survival advantage was reversed in the oldest group. In conclusion, sex and age are factors that influence the biological and clinical characteristics of colorectal cancer, affecting the metastatic pattern as well as patient survival.

## 1. Introduction

Colorectal cancer (CRC) has been estimated to be the third most common cancer worldwide in 2020, with an age-standardized incidence rate of 19.5 per 100,000, higher in men (23.4 per 100,000) than women (16.2 per 100,000) [1]. It has the second highest cancer-related mortality rate in both sexes [1]. In 2020, 43,700 new diagnoses (23,400 in men and 20,300 in women) and 21,600 deaths (11,300 in men and 10,300 in women) were expected in Italy [2]. 

The main risk factor for CRC is age: about 90% of patients are older than 50 years [3,4]. In addition, in all age groups a higher incidence has been observed in men than women, which could be attributed to behavioral or social differences: men tend to have a diet richer in red or processed meat, to be greater alcohol consumers and smokers, as well as having a greater tendency to deposit abdominal visceral fat [3,5,6,7]. All these factors have been associated with an increased risk of CRC occurrence [7,8]. However, some studies have shown that, even when controlling for these modifiable factors, differences in the incidence of the disease between men and women persist. Thus, sex appears to be a risk factor for CRC, and this has led to increased interest in studying the possible protective role of hormones [3]. It has been observed that women of childbearing age and women on hormone replacement therapy (HRT) have a lower CRC risk than men and postmenopausal women [3,6,9]. Some studies have shown that this protective role may be mediated by the estrogen receptor beta (ERβ) [9,10]. Several studies have also reported better survival in women, while others found no differences between the sexes [11,12]. The EUROCARE 5 study for CRC survival in the European population reported negligible differences between men and women [13]. 

According to numerous studies, CRC cannot be defined as a single entity but has different clinical and biological characteristics depending on its site of origin: right colon versus left colon and rectum [14,15,16]. Right colon and left colon cancers have different molecular, morphological, and clinical features that affect their management and prognosis. These features derive from their different embryological origins (middle intestine for the right colon and anterior intestine for the left colon), different blood perfusion (superior mesenteric artery for the right colon and inferior mesenteric artery for the left), and different pathways of carcinogenesis [15,16,17,18]. In fact, right colon cancer is associated more often with microsatellite instability (MSI), BRAF mutations, and CpG islet methylation, while left colon cancer more frequently presents chromosomal instability [18,19,20]. Moreover, from a morphological point of view the distribution of the mucin-associated M1 antigen is more frequent in the proximal than distal colon, which is reflected by a higher incidence of mucinous carcinomas in the right than the left colon [16,18,21]. Lesion type is also different, with a prevalence of flat-type lesions in the right colon and polypoid-type lesions in the left [14,22]. These different characteristics have an impact on the type of disease, its prognosis, response to treatment, as well as its pattern of metastasis [23,24]. 

Epidemiological studies have shown a different incidence of right and left colon cancer in relation to sex, age, and ethnicity [16]. For example, right colon cancer is more frequent in women and elderly patients, as well as in African Americans compared to Whites [14,16,18,22,25]. Tumor location also impacts survival, with right colon cancer typically having lower survival rates: it is often diagnosed at an advanced stage as it presents fewer symptoms at onset, and it tends to be less susceptible to systemic treatment than cancers of the left colon [14,18,24]. However, other studies have not shown significant differences in survival between right and left colon cancer, indicating that other factors influence prognosis [16,17,19]. One of these prognostic factors is tumor morphology: whereas most colorectal carcinomas are adenocarcinomas, 10–15% are mucinous carcinomas, a clinical entity in its own right occurring predominantly in the right colon and having a poor prognosis when metastatic [26,27]. In fact, the pattern of metastasis is determined not only by the site of onset of the tumor but also by its morphology, with adenocarcinomas metastasizing more frequently to the liver and mucinous carcinomas to the peritoneum [26]. 

While a number of studies have investigated how age or sex influence CRC and its prognosis, these factors were assessed independently of each other. Moreover, most studies looked at their influence on a particular tumor characteristic (mainly site of onset), and only a few examined the relationship to metastatic disease. The aim of our study was therefore to investigate how age and sex interact with each other in determining differences in various aspects of CRC (site of onset, tumor morphology, and pattern of metastasis) and in the survival of patients with metastases at diagnosis. To our knowledge, this is the first population-based epidemiological study that, using data from several Italian cancer registries, aims to investigate how age and sex modify the clinical characteristics of CRC, and thus its prognosis, in the Italian population. Knowledge of the epidemiological aspects of this disease may help us improve the management of patients, particularly those presenting with metastases, which may translate to an improvement of their lives. 

## 2. Materials and Methods

### 2.1. Study Design

This is a population-based study using data from four cancer registries belonging to the Italian AIRTUM network. Four population-based cancer registries participated in the study, for an annual mean observed population of 1,398,766 (2006–2017).

Each registry provided the most recent data available: Valle d’Aosta Cancer Registry (2007–2014), Pavia Cancer Registry (2006–2015), Parma Cancer Registry (2006–2015), and Ragusa–Caltanissetta Cancer Registry (2010–2017). 

Cancer registries validated their quality and completeness according to the quality checks of the International Agency for Research on Cancer (IARC) and the International Association of Cancer Registries (IACR) and they use their data for collaborative studies in cancer epidemiology research. Cases are coded according to the International Classification of Diseases for Oncology, third edition (ICD-O-3) [28]. 

Cases of invasive colon and rectal cancer with topography codes C18.0–C20.9 were selected. Patients diagnosed through death certificate only (DCO) were excluded from the analysis. The following variables were collected for each patient: sex, age at diagnosis, basis of diagnosis, tumor site, subsite and morphology, presence of metastasis at diagnosis and site of metastasis, vital status, and follow-up date.

Patients were categorized into three age groups: younger than 50 years, between 50 and 69 years, and older than 69 years. The site of tumor onset was divided into right colon (C18.0–C18.4), left colon (C18.5–C18.7 and C19.9), rectum (C20.9), and NOS (not otherwise specified, C18.8–C18.9). Morphology was grouped into four main groups as shown in Appendix A: adenocarcinoma, mucinous carcinoma, other, and NOS (Appendix A).

Passive and active monitoring of cancer cases was carried out from the date of diagnosis to at least three years, when the patients’ vital status was ascertained. The outcome variables were as follows: alive at end of follow-up; deceased including date of death from any cause; and censored due to loss or incomplete follow-up. The data were obtained through record linkage with Local Health Authority registries (listing all persons eligible for health care) and mortality registries. 

### 2.2. Statistical Analysis

A descriptive analysis of the main characteristics of the study cohort was performed. We specifically carried out detailed analyses of patients affected by cancer in the right or left colon in the adenocarcinoma or mucinous carcinoma morphology groups, and patients with metastases at diagnosis. The significance of differences between groups was analyzed by the chi-square test for categorical variables and Student’s *t*-test for continuous variables. A two-sided *p*-value less than 0.05 was considered statistically significant. 

The three-year relative survival (RS) for patients with metastatic disease was then estimated. RS is defined as the ratio of the observed survival to the expected survival in the general population of the same age and sex; it is used to correct for deaths from causes other than the cancer under investigation. RS was calculated for patients aged 0–99 years.

The Rstudio software version 1.3.1073 and STATA version 16 (strs add-on package for RS calculation) were used for the statistical analysis [29,30,31].

## 3. Results

### 3.1. Study Population

Data from 10,808 patients with CRC were collected from the four registries participating in the study. The mean age at diagnosis was 71.8 ± 12.0 years; men had a lower mean age at diagnosis (70.9 ± 11.4) than women (73.0 ± 12.6) (*p* < 0.001). The general characteristics of the study population are presented in Appendix A.

From the overall study population, we selected patients with cancer located in the right or left colon and showing adenocarcinoma or mucinous morphology (7508 patients). Mucinous carcinomas were more frequently located in the right colon (17.3% of the total vs. 7.7% in the left colon, *p* < 0.001). We found statistically significant differences in the three age groups related to sex, site of onset, tumor morphology, and metastasis at diagnosis (Table 1). The left colon was the most frequent site of onset in the younger age groups, while the oldest group showed more right colon cancer (about 54% of cases in patients >69 years vs. 40% in the two younger groups). Mucinous carcinomas were more frequent among the youngest and oldest patients than in the middle age group, while metastases at diagnosis were more frequent in the youngest cohort than in the two older groups. The middle age group had a higher percentage of male patients than the other two.

To analyze whether the observed differences were related to age, sex, or both, a separate analysis was carried out for men and women in the three age groups and another comparing men and women within the same age group (Table 2 and Table 3). Like in the overall population, all analyzed variables showed significant differences between the groups divided by sex. Among men we observed a percentage decrease in mucinous carcinomas from the youngest to the two older age groups; young men also had a higher percentage of metastases at diagnosis. Regarding the site of tumor onset, the percentages were similar between the youngest and oldest groups, while fewer right colon carcinomas were observed in the middle age group. In women we saw a progressive increase in the percentage of patients with cancer in the right colon (from 36% in the youngest group to 60% in the oldest); a similar though less pronounced trend was observed for mucinous carcinomas (11% in the two younger groups and 15% in the oldest). Like men, women in the youngest age group had the highest percentage of metastasis at diagnosis. Looking at the age groups individually, we observed that sex reached significance only for a few variables. In the <50 age group, only morphology was significant, with a higher percentage of mucinous carcinoma cases among men. In the 50–69-year-old group, only site of onset was significantly different: although the left colon was the most frequent site in both male and female patients, the percentage was much higher in men (62% vs. 55%), and the right colon showed a considerably lower percentage in women (38% vs. 45%). Lastly, in the oldest group, statistical significance was observed for both site and morphology: a right colon onset was more frequent among women (60% vs. 47%), as was mucinous carcinoma (15% vs. 11%).

### 3.2. Metastasis

The cohort of metastatic patients was selected from the overall study population and consisted of 2038 patients: 1092 men (53.6%) and 946 women (46.4%). The mean age at diagnosis was 69.6 ± 12.1 years in men and 71.7 ± 12.9 years in women (*p* < 0.001). Looking at differences in disease type within the metastatic cohort, we observed statistically significant differences in sex, site of onset, and tumor morphology between the three age groups (Table 4). In the two younger groups men outnumbered women, while in the oldest group the two sexes tended to be represented equally. With regard to the site of onset, the left colon decreased in frequency from the youngest to the oldest group, while the right colon showed an opposite trend. Adenocarcinoma morphology was the most frequent in all three age groups; mucinous carcinoma showed a decrease in frequency from the youngest to the oldest group, though the latter also showed a high percentage (17.8%) of cases with NOS morphology. 

Of the 2038 patients with metastasic disease at diagnosis, the site of metastasis was known in 74% (1511 patients). The distribution of metastatic sites in men and women of each age group is shown in the charts of Figure 1. The most frequent site of metastasis across age groups and sexes was the liver, although the mean percentage was lower in women (47%) than in men (56%). The metastatic pattern was more homogeneous in men of all ages; women showed different metastatic patterns in the three age groups, with more frequent involvement of the peritoneum and other sites compared to men.

We then analyzed the most frequent sites of metastasis (liver, peritoneum, and lung) by sex and age. While statistically significant differences within age groups were observed only for the liver and lung (Table 5), the charts of Figure 2 show the differences observed in metastatic sites by sex and age. About 70% of male patients of each age group had liver metastases at diagnosis, but in women there were significant differences between age groups. In the youngest and oldest female groups, statistically significant differences were also observed with respect to men of the same age. The peritoneum was a more frequent site of metastasis in women than in men, although the difference between the two sexes was significant only in the oldest group. Metastases to the lung were more frequent in men of the two younger groups and in women of the oldest group. No significant difference between the sexes was observed in the singular age groups, but between women of different age groups the difference was statistically significant. Lastly, we looked for differences in the presence of multiple sites of metastasis at diagnosis, regardless of the site involved: no significant differences were detected by either sex or age, though the percentage of multiple sites was higher in men and women of the youngest age group.

### 3.3. Survival 

We analyzed three-year RS in the cohort of metastatic patients in relation to sex and age. Men had a survival disadvantage in the youngest age group, but the distance between the curves narrowed from the middle to the oldest age group, where the disadvantage seemed to shift to women (Figure 3). Survival by sex and age was then analyzed in relation to tumor site (right vs. left colon; Appendix A) and morphology (adenocarcinoma vs. mucinous carcinoma; Appendix A). The right colon was associated with lower survival rates in both women and men in the two younger groups, while in the oldest group the curves tended to overlap. In terms of morphology, mucinous carcinoma showed lower survival rates in both sexes in the younger groups, while in the oldest group the curves again tended to overlap. 

Lastly, we performed a survival analysis for the most frequent sites of metastasis (Figure 4). Liver metastases affected survival differently between the two sexes: in men the presence or absence of liver metastases did not seem to change the survival curve, whereas women of all age groups showed a worse survival when liver metastases were present. A similar trend was observed for lung metastases; however, middle-aged and older women with lung metastases had better survival than women without lung metastases. The peritoneum was very similar in both sexes: survival was worse in the presence of peritoneal metastases, especially in the youngest age group. Lastly, in both sexes survival was worse in patients with multiple metastases than in those with a single site of metastasis. In the youngest age group, women with a single site of metastasis had better survival than their male counterparts, whose survival curve was superimposable to that of male patients with multiple metastases. 

## 4. Discussion

From our analyses it can be observed that both sex and age influence the biological and clinical features of CRC, affecting the pattern of metastasis as well as patient survival. The first evidence was that women had a significantly higher mean age at diagnosis than men both in the overall study population and the metastatic cohort. 

Older patients showed a higher percentage of right colon cancer than those in the younger age groups, as previously reported by other studies [14,25,32]. Sex- and age-specific analysis, however, showed that the increase in right colon cancers in the older age groups was much more marked in women: female patients of the youngest group had a significantly lower percentage of right colon cancer than women in the older age groups (36% vs. 45% vs. 60%). They also had a lower percentage of right colon cancer than younger males (36% vs. 45%), but the difference was not statistically significant, probably because of the small number of cases. In the other two age groups women showed a significantly higher percentage of cases in the right colon than their male counterparts. 

Several studies found a higher percentage of right colon cancer in women than men [19,23,32]. However, in these studies age was considered separately from sex. Our study, which analyzed sex in the three age groups mentioned, showed that in women the increase in right colon cancer from the youngest to the oldest group was much greater than in men, although the percentage of right colon cancers exceeded that of left colon cancers only in the oldest group. Our data suggest that women in the youngest group may have a comparable risk to men of the same age (although the female percentage in our study was smaller, but not significantly so) and that female sex therefore increases the risk of right colon cancer only in the older age groups. Furthermore, the progressive increase with age in women could indicate the disappearance of a protective factor, namely sex hormones, as also suggested by other studies. The risk of women in the middle age group might be affected by the use (or not) of HRT. However, the lack of data regarding menopause and HRT use does not allow us to assess this effect in our study population. 

Regarding tumor morphology, it is known that mucinous carcinoma tends to have a higher incidence in the right colon than the left [16,18,21]. Some studies have shown a higher percentage of mucinous tumors in younger patients (21% versus 10–15% in patients of all ages) [33,34]. In our study, younger patients showed a higher percentage of mucinous carcinomas than the other two age groups, but the difference turned out to be solely attributable to male patients having a higher mucinous carcinoma percentage in that age group. In addition, an opposite trend was observed in men and women, with men showing a decrease in mucinous carcinomas from the youngest to the two older groups, while in women there was an increase in cases from the youngest to the oldest group. Another study showed that women have a higher proportion of mucinous carcinomas than men [35]. Again, our analysis by sex and age showed that this is true only for the >69 age group (11% in men vs. 15% in women, *p* < 0.001); in the middle age group there was no difference between men and women, while in the youngest group men showed a significantly higher percentage of mucinous carcinomas (22% vs. 11%). 

It has been reported that about 20% of patients with CRC have metastases at diagnosis [26,36]. In our analysis of patients with right or left colon cancer and adenocarcinoma or mucinous carcinoma morphology, there were 1244 patients with metastasis at diagnosis (about 17% of the total). Qiu et al. in their study of SEER data from the U.S. found the percentage of stage IV patients to be 18.1%, similar to ours [36]. We observed significant differences between age groups in the overall study population and in the groups divided by sex, while no differences were observed between the sexes in the same age groups. This could mean that age is a more important determinant than sex for the presence of metastasis at diagnosis, with younger people being more frequently affected than older people. Colon cancer screening might play a role here, as cancer is being detected at an earlier stage in the older age group thanks to screening. However, it should be taken into account that in our study more patients in the older age groups lacked information regarding metastasis at diagnosis than those in the youngest group (25% vs. 24% vs. 19%).

Analyses conducted on patients with right or left colon carcinoma and adenocarcinoma or mucinous morphology allowed us to hypothesize that some differences are due solely to age, such as the higher frequency of metastases at diagnosis in young patients regardless of sex. Many other observed differences, however, seem to depend on both age and sex, such as those related to the site of tumor onset and morphology.

### Metastasis

Among patients with metastases at diagnosis, the liver was the most frequent site in women and men alike. Some variability was observed especially within the female age groups: in the youngest group, liver metastases amounted to just 36% of the total, while the peritoneum (18%) and other sites (20%) showed a relatively high frequency. Notably, "other" refers to sites other than the liver, peritoneum and lung: the proportion is particularly high in women, and in the two younger age groups this is mainly due to involvement of the female genital tract (44% in the youngest, 40% in the middle and 19% in the oldest group). Metastases to other sites in men were mainly to distant lymph nodes (mean 19% in the three age groups) and bone (particularly in the two older groups, 22% and 30%, respectively). The metastatic pattern observed in men appeared more homogeneous than in women, who showed different metastatic patterns in the three age groups.

The most common sites of CRC metastasis reported in the literature are the liver and lung [36]. Our study showed that, if only age is taken into account, the liver was significantly more frequently involved in the middle age group, while lung metastases increased significantly from the youngest to the oldest patients (10% vs. 11% vs. 15%). When patients were divided by sex, these differences were maintained only in women: lung metastases rose from 7% in the youngest to 16% in the oldest group, while liver metastases occurred in 51%, 70%, and 58%, respectively, in the three age groups. Within age groups significant differences were observed for the liver in the youngest and oldest groups, with a lower frequency among women, and for the peritoneum, where women in the oldest group showed a higher frequency. No significant differences were observed for multiple sites of metastasis, although there was a trend towards higher percentages in young patients. One study found that liver metastases were more frequent in younger than in older patients in the five years after diagnosis [37]. 

Many studies have analyzed patterns of metastasis in relation to the site of tumor onset and morphology. For example, it was found that left colon cancer metastasizes more often to the liver and lung, and right colon cancer to the peritoneum and brain [17,37]. In terms of morphology, adenocarcinomas seem to metastasize more to the liver and mucinous carcinomas to the peritoneum; metastases to ovary, subcutaneous tissues, and skin are also more common in mucinous carcinomas [26]. The differences by sex and age observed in the present study might, therefore, be explained by differences related to morphology or site of onset. However, the particular trends we observed between men and women, combined with the differences observed in the overall study population, could also suggest the involvement of sex and age in defining the metastatic pattern, especially in women. 

Survival analysis in the metastatic cohort showed better RS in the youngest age group for women, whereas in the older age groups the female advantage lessened to the point of reversing in the oldest group. A survival advantage in younger women over men of the same age was also observed by other studies, but not in older patients [5,6,11]. With regard to site of onset and morphology, in the metastatic cohort survival was worse for right versus left colon primaries and for mucinous carcinomas versus adenocarcinomas in both sexes and all age groups but the oldest, in which the curves tended to overlap for both tumor site and morphology. Some studies have shown a worse prognosis for right versus left colon carcinomas also because of the higher proportion of mucinous carcinomas in the right colon, as these are associated with a poor prognosis especially in metastatic disease [24,26,38]. Other studies, however, have not shown significant differences in survival between right and left colon cancer [16,17,19].

Survival analysis by site of metastasis showed differences in age- and sex-specific curves according to different sites. Peritoneal metastases were reported in the literature to have a poor prognosis, while lung metastases seem to be associated with better survival than liver metastases [37,39]. In our analysis, at all ages and for both sexes, a worse prognosis was observed for patients with peritoneal metastases, particularly in the youngest group. Women with lung metastases showed better survival than those without lung metastases in the two older age groups, while in men the survival curves for liver metastases and lung metastases overlapped. Women without liver involvement had better survival in all age groups. Sex, therefore, could play a role in patient survival depending also on the site of metastatic involvement. 

Several studies have suggested a protective role of female sex hormones in the development of CRC: women who have used oral contraceptives are at lower risk of developing CRC than those who have not, just as HRT appears to have a protective role in postmenopausal women [4,9]. A protective mechanism has been proposed involving mainly the estrogen receptor ERβ, which has been found to be involved in the reduction of colorectal adenomatous polyps and the modulation of some CRC pathways [9,10]. Polymorphisms have been observed also in the gene encoding the EGFR receptor; these have been positively associated with CRC survival in women and negatively in men, highlighting the possibility that the interaction between EGFR and ERβ signaling underlies the differences between the two sexes [10]. 

Our study has some limitations, mainly due to a lack of information on certain variables, such as disease stage, treatment, molecular and genetic profile, and incomplete information on others, including site of onset (NOS 7%), morphology (NOS 7%), and site of metastasis (known in only 74% of metastatic patients), preventing us from using the data of some patients. In addition, the lack of information regarding the use of oral contraceptives or HRT in women did not allow us to analyze their relationship with disease characteristics and prognosis. A strength of the study, however, is the use of data from population-based registries, which, unlike hospital-based observational data whose strength is the clinical detail, allowed us to provide a general view of the disease in the source population in all its heterogeneity and possibly without any selection bias.

The debate about the best age for screening initiation based on individual risk factors (family history) and also in relation to sex was raised in the study by Wernly et al., which showed that male sex was the most important risk factor for CRC [40]. This could prompt a discussion on the possibility of reducing the screening age for men in an attempt to forestall the onset of the disease in what appears to be the most at-risk population.

## 5. Conclusions

Colorectal cancer is among the cancers with the highest incidence in the world population. Through data from cancer registries, it is possible to assess how biological factors including sex and age influence the biological and clinical characteristics of the disease. Our analysis found interesting differences between men and women that were not always consistent in the three age groups, sometimes showing an opposite trend in the two sexes from the youngest to the oldest group. As hypothesized by other studies, the differences between men and women might be attributable to a protective role of female hormones against CRC.

Fully understanding the factors that affect the characteristics and prognosis of such a widespread cancer may allow the development of appropriate screening and prevention strategies based on sex and age. This could contribute to the earlier diagnosis of CRC and the development of clinical management pathways aimed at improving patients’ quality of life and survival.

## Figures and Tables

**Figure 1 cancers-15-00803-f001:**
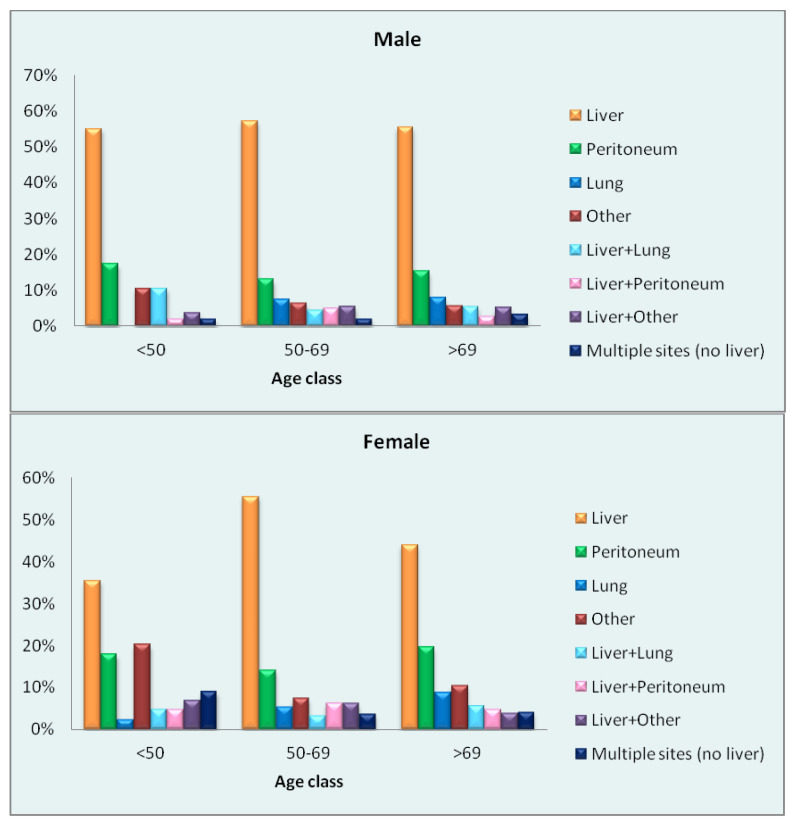
Percentage distribution of sites of metastasis by sex and age in patients with metastatic disease at diagnosis.

**Figure 2 cancers-15-00803-f002:**
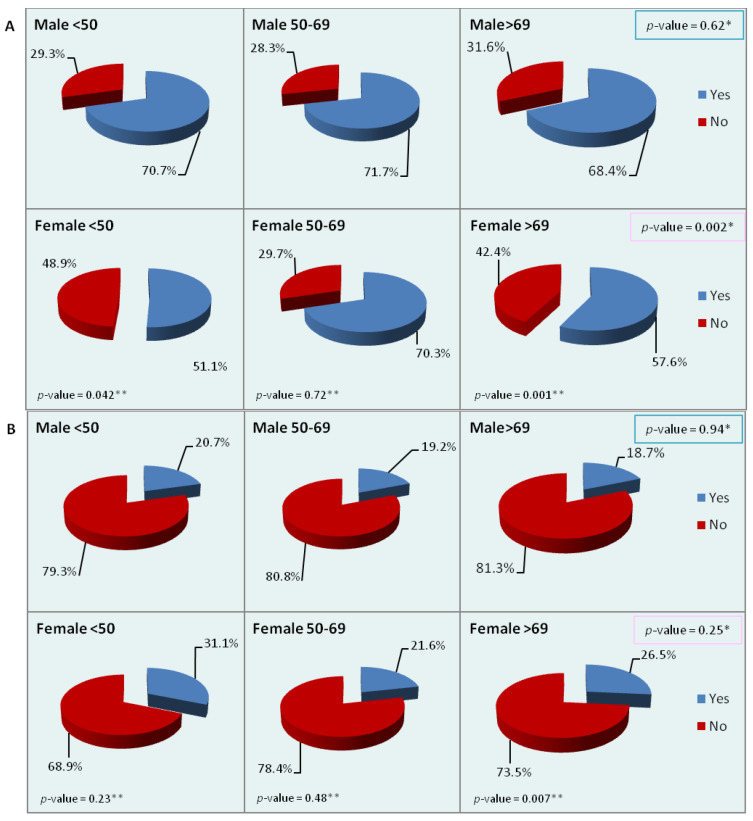
Percentage distribution of the main sites of metastasis by sex and age ((**A**): liver; (**B**): peritoneum; (**C**): lung; (**D**): multiple sites). *p*-values refer to the chi-square test for between-group differences.

**Figure 3 cancers-15-00803-f003:**
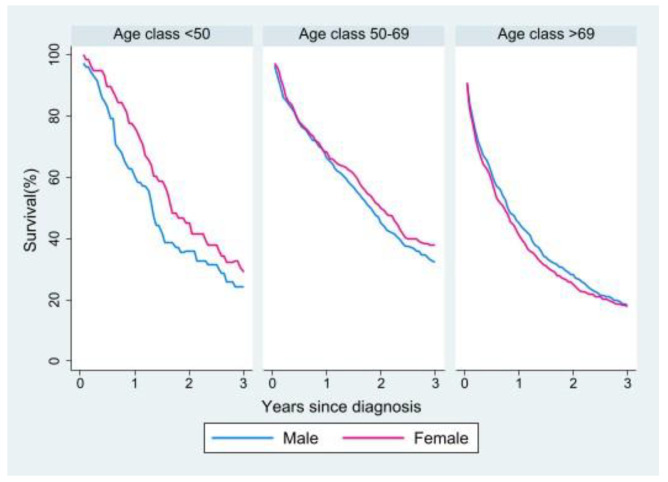
Relative survival of patients with metastasis at diagnosis by age and sex.

**Figure 4 cancers-15-00803-f004:**
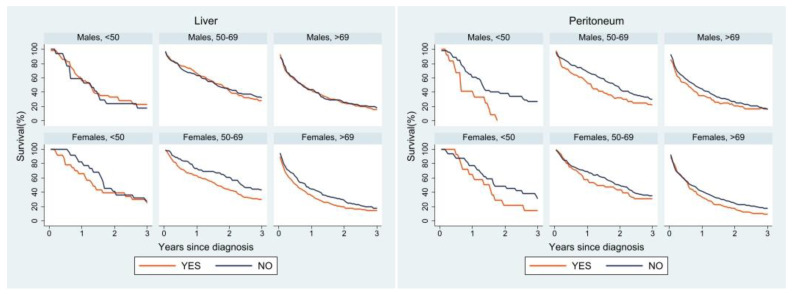
Relative survival of patients with metastasis at diagnosis by age, sex, and site of metastasis.

**Table 1 cancers-15-00803-t001:** Absolute and percentage distribution by age of the main variables in patients with right or left colon cancer and adenocarcinoma or mucinous morphology. *p*-values refer to the chi-square test for between-group differences.

*N* = 7508				
Age (Years)	<50 *n* (%)	50–69 *n* (%)	>69 *n* (%)	*p*-Value
*N*	297 (4.0)	2627 (35.0)	4584 (61.0)	
Sex				
Male	146 (49.2)	1579 (60.1)	2357 (51.4)	<0.001
Female	151 (50.8)	1048 (39.9)	2227 (48.6)	
Site				
Left colon	178 (59.9)	1566 (59.6)	2122 (46.3)	<0.001
Right colon	119 (40.1)	1061 (40.4)	2462 (53.7)	
Metastasis				
No	158 (53.2)	1559 (59.3)	2724 (59.4)	<0.001
Yes	82 (27.6)	446 (17.0)	716 (15.6)	
Unknown	57 (19.2)	622 (23.7)	1144 (25.0)	
Morphology				
Adenocarcinoma	248 (83.5)	2334 (88.8)	3997 (87.2)	0.011
Mucinous carcinoma	49 (16.5)	293 (11.2)	587 (12.8)	

**Table 2 cancers-15-00803-t002:** Absolute and percentage distribution by sex and age of the main variables in patients with right or left colon cancer and adenocarcinoma or mucinous carcinoma morphology. *p*-values refer to the chi-square test for between-group differences.

	MALE				FEMALE			
Age (Years)	<50 *n* (%)	50–69 *n* (%)	>69 *n* (%)	*p*-Value	<50 *n* (%)	50–69 *n* (%)	>69 *n* (%)	*p*-Value
*N*	146 (3.6)	1579 (38.7)	2357 (57.7)		151 (4.4)	1048 (30.6)	2227 (65.0)	
Site								
Left colon	81 (55.5)	985 (62.4)	1241 (52.7)	<0.001	97 (64.2)	581 (55.4)	881 (39.6)	<0.001
Right colon	65 (44.5)	594 (37.6)	1116 (47.3)		54 (35.8)	467 (44.6)	1346 (60.4)	
Metastasis								
No	75 (51.4)	941 (59.6)	1404 (59.6)	<0.001	83 (54.9)	618 (59.0)	1320 (59.3)	0.02
Yes	44 (30.1)	257 (16.3)	364 (15.4)		38 (25.2)	189 (18.0)	352 (15.8)	
Unknown	27 (18.5)	381 (24.1)	589 (25.0)		30 (19.9)	241 (23.0)	555 (24.9)	
Morphology								
Adenocarcinoma	114 (78.1)	1408 (89.2)	2103 (89.2)	<0.001	134 (88.7)	926 (88.4)	1894 (85.0)	0.02
Mucinous carcinoma	32 (21.9)	171 (10.8)	254 (10.8)		17 (11.3)	122 (11.6)	333 (15.0)	

**Table 3 cancers-15-00803-t003:** Absolute and percentage distribution by age and sex of the main variables in patients with right or left colon cancer and adenocarcinoma or mucinous carcinoma morphology. *p*-values refer to the chi-square test for between-group differences.

Age (Years)	<50			50–69			>69		
	Male *n* (%)	Female *n* (%)	*p*-Value	Male *n* (%)	Female *n* (%)	*p*-Value	Male *n* (%)	Female *n* (%)	*p*-Value
*N*	146 (49.2)	151 (50.8)	0.8	1579 (60.1)	1048 (39.9)	<0.001	2357 (51.4)	2227 (48.6)	0.05
Site									
Left colon	81 (55.5)	97 (64.2)	0.12	985 (62.4)	581 (55.4)	<0.001	1241 (52.7)	881 (39.6)	<0.001
Right colon	65 (44.5)	54 (35.8)		594 (37.6)	467 (44.6)		1116 (47.3)	1346 (60.4)	
Metastasis									
No	75 (51.4)	83 (54.9)	0.63	941 (59.6)	618 (59.0)	0.47	1404 (59.6)	1320 (59.3)	0.94
Yes	44 (30.1)	38 (25.2)		257 (16.3)	189 (18.0)		364 (15.4)	352 (15.8)	
Unknown	27 (18.5)	30 (19.9)		381 (24.1)	241 (23.0)		589 (25.0)	555 (24.9)	
Morphology									
Adenocarcinoma	114 (78.1)	134 (88.7)	0.01	1408 (89.2)	926 (88.4)	0.52	2103 (89.2)	1894 (85.0)	<0.001
Mucinous carcinoma	32 (21.9)	17 (11.3)		171 (10.8)	122 (11.6)		254 (10.8)	333 (15.0)	

**Table 4 cancers-15-00803-t004:** Absolute and percentage distribution by age of the main variables in patients with metastasis at diagnosis. *p*-values refer to the chi-square test for between-group differences.

*N* = 2038				
Age (Years)	<50 *n* (%)	50–69 *n* (%)	>69 *n* (%)	*p*-Value
*N*	130 (6.4)	717 (35.2)	1191 (58.4)	
Sex				
Male	72 (55.4)	415 (57.9)	605 (50.8)	0.01
Female	58 (44.6)	302 (42.1)	586 (49.2)	
Site				
Left colon	54 (41.5)	277 (38.6)	409 (34.3)	<0.001
Right colon	34 (26.2)	201 (28.1)	437 (36.7)	
Rectum	26 (20.0)	168 (23.4)	205 (17.2)	
NOS	16 (12.3)	71 (9.9)	140 (11.8)	
Morphology				
Adenocarcinoma	94 (72.3)	560 (78.1)	842 (70.7)	<0.001
Mucinous carcinoma	25 (19.2)	86 (12.0)	122 (10.2)	
Other	3 (2.3)	16 (2.2)	15 (1.3)	
NOS	8 (6.2)	55 (7.7)	212 (17.8)	

**Table 5 cancers-15-00803-t005:** Absolute and percentage distribution by age of the main sites of metastasis at diagnosis. *p*-values refer to the chi-square test for between-group differences.

*N* = 1511				
Age (Years)	<50 *n* (%)	50–69 *n* (%)	>69 *n* (%)	*p*-Value
*N*	103 (6.8)	554 (36.7)	854 (56.5)	
Liver *				
Yes	64 (62.1)	394 (71.1)	538 (63.0)	0.005
No	39 (37.9)	160 (28.9)	316 (37.0)	
Peritoneum *				
Yes	26 (25.2)	112 (20.2)	193 (22.6)	0.40
No	77 (74.8)	442 (79.8)	661 (77.4)	
Lung *				
Yes	10 (9.7)	60 (10.8)	130 (15.2)	0.03
No	93 (90.3)	494 (89.2)	724 (84.8)	
				0.65
Multiple sites				
Yes	21 (20.4)	95 (17.1)	143 (16.7)	
No	82 (79.6)	459 (82.9)	711 (83.3)	

* Single site or in multiple sites.

## Data Availability

The data on colon cancer cases used in this study were provided by cancer registries affiliated to AIRTUM and cannot be made freely available.

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
