# Peer review of "Influence of Sex and Age on Site of Onset, Morphology, and Site of Metastasis in Colorectal Cancer: A Population-Based Study on Data from Four Italian Cancer Registries"

_cancers, 2023, doi:10.3390/cancers15030803_

Round 1

Reviewer 1 Report

Thank you for the opportunity to review this manuscript by Perotti for Cancers.

The authors retrospectively reviewed data on sex and age in a population-based cohort of patients with colorectal cancer. They gathered more than 10,000 patients. They found some differences between sexes and between age groups. A severe limitation of these kind of studies is that many variables are lacking (i.e. MMR status). Moreover, the authors were specifically interested in metastatic patients. In an important number of patients it was unknown whether or not they had metastases and in the group of patients with metastases the site was often unknown. These are serious limitations which avoid to draw strong conclusions. I have outlined several questions below.

Abstract: 

‘Women showed significant differences in right colon cases from the youngest to 38 oldest age group (36% vs 45% vs 60%)’ probably you mean the proportion of female patients per age catgeory. Be precise.

Introduction: this should be more tot he point.I’d suggest to reduce the text by a third.

Methods: the methods section should be extended. How many people live inside the area covered by these four registries? How reliable is the data? How was the side of metastases assessed? Biopsy? How did the authors classify double tumors? Did they collect MMR status? If available, did they use the clinical or pathological TNM? Why did they group left colon and rectal together? I’m confused is rectal cancer included and grouped under left colon cancer or excluded? The cohorts are not overlapping completely. I would select patients from completely overlapping years only, as diagnostics and guidelines may change.

Was there population-based screening in one of the regions? If so, what kind of screening?

How did the authors account for multiple testing.

Results:

Why were 2300 patients excluded?

The proportion of unknown metastases is huge. In up to 25% of patients with known metastases the site is missing. This has severe impact on the conclusions that can be drawn.

In Table 2,3 and 4 the authors only look univariably for differences. Peritoneal metastases occurred more frequent in older women, but they also had mucinous tumors more frequently. 

The authors looked at multiple sites of metastase. I would argue that liver only metastases and peritoneal only metastases are most relevant because patients with these metastases may still be treated with curative intent.

I miss numbers at risk below the survival curves. Again, the authors address survival only univariably. As s many factors are different this does not make sense

Discussion:

Many results are repeated here. This is unnecessary.

There is an in conclusion paragraph and a formal conclusion paragraph. One of them is redundant.

“In conclusion, the influence of sex and age on the biological and clinical characteristics of CRC pose important questions regarding the management of patients in relation to their biological characteristics “ why? It is not an on or off phenomenon. I cannot see how you would adjust screening based on these data.

Reviewer 2 Report

The authors of the manuscript “Influence of sex and age on site of onset, ….” performed an investigation about how age and sex modify the clinical characteristics of colorectal cancer (CRC), and its prognosis in the Italian population.

The significant data collection from cancer registries belonging AIRTUM network allowed the authors the categorization of 10,808 CRC-bearing patients into three age groups.

This epidemiological study highlighted an interesting involvement of the non-modifiable factors (sex and age) in CRC development; in particular, sex and age influence the site of onset, morphology, and metastatic pattern of CRC, affecting the pattern of metastasis and patient survival.

The authors identified some limitations in this study: lack of information or incomplete information. However, the analyses had been carried out by using data derived from population-based registries, without selection bias.

Understanding and knowing the mechanisms and factors underlying the development of cancer allows to design appropriate screening and prevention strategies based on sex and age, contributing to the earlier diagnosis of CRC.

Minor comments:

The authors cited some observations related to sex hormones (through ERbeta), HRT adoption, EGFR polymorphisms, behavioral or social differences, etc.
Please, if possible, add, in the discussion section, information on factors and mechanisms promoting CRC -related to sex and age differences- and/or site of onset (left or right colon) and site of metastasis.

Regarding the attention to the development of appropriate screening and prevention strategies based on sex and age, improving the quality of life and survival of patients, to contribute to early diagnosis and the development of clinical management pathways:
P
lease, in the discussion section, could you suggest/propose some models?

Reviewer 3 Report

In the manuscript, Perotti et al., studied a population based analysis that how age and sex factors changes the clinical  characteristics of colorectal cancer in the Italian population. Here Author used four different registry from airtum network such as Valle d'Aosta Cancer Registry, Pavia Cancer Registry, Parma Cancer Registry, and Ragusa–Caltanissetta Cancer Registry. Patients were grouped into three category that are below 50 years,  50 to 69 years and above 69 years. very broad range of age group have chosen. they would have done it in 5year interval instead of 19years gap in mid age group.  Basic descriptive analysis were performed in detail with available data.   lack of patient disease information such as mutation, genetic instability, somatic and so on.  Over all the manuscript is well written and each section is aptly described with necessary details. 
